# Accuracy of perceived glaucoma risk by patients in a clinical setting

Chiun-Ho Hou[1,2,3,4], Jiahn-Shing Lee[1,4], Ken-Kuo Lin[1,4], Laura Liu[1,4], Yung-Sung Lee[1,4], Christy Pu[2]*

1 Department of Ophthalmology, Linkou Chang Gung Memorial Hospital, Taoyuan, Taiwan, 2 Institute of Public Health, School of Medicine, National Yang Ming Chiao Tung University, Tainan, Taiwan, 3 Department of Ophthalmology, Xiaman Chang Gung, Xiaman, People's Republic of China, 4 Department of Medicine, College of Medicine, Chang Gung University, Taoyuan, Taiwan

* cypu@nycu.edu.tw

## Abstract

### Objective

To determine whether patients attending the ophthalmology department underestimate their glaucoma risks.

### Method

We conducted a cross-sectional survey with a final study population of 1203 individuals from two medical centers in Taiwan during January 1–June 30, 2019. The "High concern" group was defined as the set of patients who rated themselves as having low risk but who had been rated by physicians as having medium or high risk of developing glaucoma over the next year.

### Results

Approximately 12% of the respondents belonged to the "High concern" group. For those with education at the college level or higher, the interaction term was estimated to be 0.294 (95% CI = 0.136–0.634). Marginal effect calculations revealed significant sex-based differences in the effect of knowledge at specific age intervals.

### Conclusions

A considerable proportion of patients attending the ophthalmology department underestimate their glaucoma risks. Misjudgment of glaucoma risks can lead to delays in seeking of medical attention. Glaucoma education should be designed according to each patient's education level and sex, as its effect is not consistent across different education and sex clusters.

**Data Availability Statement:** Data cannot be shared publicly because of IRB restrictions by the IRB of National Yang-Ming University. According to the IRB restriction, all individuals who wishes to analyze the data must be specified in the IRB

application. Thus, we can not share this dataset online. If a person who was not specified in the IRB application wishes to analyze the data, he/she must obtain approval from the above named IRB. Contact information for ethics committee: email: irb@ym.edu.tw Tel: +886-2-2823-9753.

**Funding:** This study was supported by Ministry of Science and Technology, Taiwan (Grant number: 109-2314-B-010 -049 -MY2). Grant recipient: C Pu. URL: https://www.most.gov.tw/?l=en The funders had no role in study design, data collection and analysis, decision to publish, or preparation of the manuscript.

**Competing interests:** The authors have declared that no competing interests exist.

## Introduction

Late presentation for glaucoma is a major risk factor for blindness [1]. Patients often go blind due to glaucoma without ever having been diagnosed [2], and this problem has not improved over the past 2 decades [3]. The Baltimore Eye survey showed that approximately 50% of the people in the United States with optic nerve damage from primary open-angle glaucoma were unaware that they have the condition [4]. A study conducted on the Hispanic population in the US found that up to 62% of the study subjects with open-angle glaucoma had not been aware of their condition before they were screened; the prevalence of undiagnosed glaucoma increases with age [5].

Improving awareness of symptoms of glaucoma, signs of glaucoma, and knowledge regarding glaucoma, however, does not guarantee prompt diagnosis, because an undiagnosed patient may have high awareness and knowledge of glaucoma, but may not perceive he or she is at glaucoma risk. To initiate a medical event, the patient must first suspect the possibility of a medical condition.

Perceived risk has been studied for other diseases but has rarely been studied in ophthalmology. For other diseases [6], found that a perceived risk of falls is useful in assessing the risk of falls for elderly individuals. Similar studies have been performed on cardiovascular diseases, wherein the perceived risk is positively associated with the actual event [7]. Perceived breast cancer risk has also been found to have mediated the negative effect of pain on distress for post-surgery breast cancer survivors, as well as unaffected women [8].

Despite various studies that have been performed to assess perceived risks, two shortcomings still remain to be addressed. First, no such study has been undertaken for glaucoma. Glaucoma differs from many chronic diseases in that glaucoma often progresses without obvious symptoms, and the progression of the condition often remains unnoticed. Thus, it may not be safe for a patient to compare self-perceived risks with actual events because perceived high risk based on symptoms would lead to late presentation for glaucoma. Second, previous studies on the perceived risk have not evaluated whether such perceptions are valid.

Studies on the role of education level and glaucoma are scarce. Socioeconomic status such as education can affect glaucoma prevalence independent of any hereditary predisposition [9]. A study by Fraser et al. [10] has demonstrated that educational deprivation is a key determinant of late presentation for glaucoma as socioeconomic status influences awareness of the disease and the need for regular eye-sight testing. Hoevenaars et al. [11] found that knowledge of glaucoma and its treatment is positively associated with socioeconomic status. Oh et al. [12] found that the diagnosed prevalence of glaucoma decreased with education. The current study contributes to the literature in two ways. First, we examine patients' perception of glaucoma and compare with risks evaluated by their ophthalmologists, and second, we evaluate whether the effect of knowledge on glaucoma and education play a role in this risk-evaluation discrepancy.

## Methods

We conducted a cross-sectional survey from January 1 to June 30, 2019. The questionnaires were self-administered, in paper form, by patients who visited the department of ophthalmology at two medical centers under one hospital chain in the Northern part of Taiwan (Chang Gung Memorial Hospital). Patients older than or equal to 20, willing to participate, and were able to complete the questionnaire were included. The survey was self-administered at the site, with trained assistants standing ready to answer any questions from the respondent. The questionnaire was developed by the authors (see S1 and S2 Files for the questionnaire). The questionnaire consisted of sections on the demographic and socioeconomic status of the

respondents, on knowledge of causes, symptoms, and signs of glaucoma (explained in detail later), and a section on physician ratings. The questionnaire was reviewed by six ophthalmologists and two public health experts. Approximately 15 patients from the pre-test were asked to fill out the questionnaire again upon their next visits for reliability testing. The questionnaire was then modified accordingly. The National Yang Ming University Institutional Review Board (IRB) approved the study (IRB number: YM107106F). Written informed consents were obtained from all participants.

A total of 1203 valid questionnaires were collected. We were not able to calculate an exact response rate because a patient can only participate in the survey once, and hence patients with repeated visits during the study period were excluded if they had already been required to fill out the questionnaire during a previous visit. A response rate that used all patients as the denominator thus would significantly underestimate the response rate.

## Self-assessed and physician-assessed risk of glaucoma

The respondent was asked the question "Do you think you are at a risk of developing glaucoma over the next year?" The respondents chose from five responses, namely (1) High risk, (2) Medium risk, (3) Low risk, (4) I have no idea, and (5) I already have glaucoma.

An ophthalmologist evaluated whether he/she considered the respondent to be at a risk of developing glaucoma over the next 1 year based on the same five responses listed previously. We asked the physicians to rate the patient's risk based on his/her medical examination results (visual field and optical coherent tomography) and the following factors: (1) age over 40 years; (2) family members with glaucoma; (3) high ocular pressure; (4) severe farsightedness or nearsightedness; (5) past ocular injury; (6) long-term steroid medication; (7) has corneas that are thin in the center; (8) thinning of the optic nerve, and (9) diabetes and high blood pressure. The physician was asked to tick on a paper form whether the aforementioned nine conditions were present for the patient and to provide an overall assessment based on these conditions.

We can define the "High concern group" as consisting of patients who rate themselves as having low risk or having no idea of his/her risk, whereas his/her physician would rate him/her as having medium or high risk. For comparison purposes, we also included a question asking the patients to rate their risks for future development of glaucoma compared with other people of similar age.

## Education and glaucoma knowledge

Education is self-reported by the respondent and is a categorical variable with the following four categories: (1) Primary school or illiterate; (2) Junior high school; (3) Senior high school, and (4) College and above. A series of questions on glaucoma knowledge were asked. The selection of the knowledge questions was based on standard literature. The answers must be selected from three options; (1) True, (2) False, and (3) Don't know. These questions were as follows:

1. Glaucoma can be completely cured

2. Cataracts can lead to glaucoma

3. Over-use of the eyes can lead to glaucoma

4. People with no family history have lower chances of getting glaucoma

5. Early-stage glaucoma does not have symptoms

6. In general, an ocular pressure lower than 30 mmHg is considered normal.

We subsequently included a series of questions based on glaucoma symptoms:

1. Blurred vision

2. Pain in the eye

3. Halo around light

4. Nausea

5. Headache

## Statistical analysis

Because our sample was drawn from a nonprobability sample, we employed the quasirandomization weighting that had been proposed by Valliant [13]. This weighting process estimates pseudo-inclusion probabilities of each individual using variables that are available in both the survey and national representative data set (the reference data set). Because this method required individual data from the reference survey, we used 2013 National Health Insurance (NHI) claims data that included one million randomly selected individuals in Taiwan. The year 2013 was the most up-to-date available year that allowed us to obtain data and to match them to our survey data. For efficient estimation, we randomly selected 50,000 individuals from the data set. The common variables available for our survey and the NHI data were age, sex, and place of residence. Because our survey was not intended to represent the general population (the individuals were patients selected on site), we eliminated individuals who had not visited the ophthalmology department from the reference data set. To test the consistency of this weight assignment, we repeated the weight calculation 10 times with independent random selection of 50,000 individuals and recalculated the weight for each random selection. The overall correlation between each of these 10 calculations was greater than 0.95. To further ensure our weight was not biased due to variation in weighting methods, we executed model-based weighting using sub-population totals from the same reference data set. The correlation of the weights obtained between this method and the pseudo-inclusion probabilities methods still remained greater than 0.95. For a thorough discussion on the significance of weighting and methods of weighting as previously described, please refer to Valliant [13].

For the glaucoma knowledge questions, the proportions of respondents that reported the correct answers were recorded. We then assigned one point for each correct answer, summed the scores, and treated the variable as a continuous variable.

Furthermore, we designed a logistic regression model using a model-building process [14]. First, we conducted an exploratory bivariate analysis to determine relevant variables. These variables were required be scientifically relevant to the outcome variable (risk-evaluation discrepancy). Subsequently, we retained variables with a statistical significance of $p < 0.25$ based on the bivariate analysis. Wald tests for multiple coefficients were subsequently applied to remove redundant variables to make the model parsimonious. Because our variables of interest were education and knowledge score, we tested interactions between these two variables, as well as interactions for these two variables along with age and sex. Subsequently, we repeated the estimation by removing non-significant interactions. We performed a goodness-of-fit test at each stage to check model fit. A model was not considered fit unless the goodness-of-fit test was passed. The final model was developed using the aforementioned process. Based on the final model, we calculated the marginal effect of the main variables of interest.

## Results

Table 1 shows the proportions of responses for each category for patients and physicians. A total of 141 individuals were found to fit the definition of the "High concern" group. Most

**Table 1. Self-rated glaucoma risk and physician-rated glaucoma risk, unweighted.**

| Responses by patient/physician | Self-rated | | Physician-rated | | P value |
|---|---|---|---|---|---|
| (1) High risk | 22 | 1.83 | 39 | 3.24 | <0.0001 |
| (2) Medium risk | 29 | 2.41 | 125 | 10.39 | |
| (3) Low risk | 359 | 29.84 | 762 | 63.34 | |
| (4) Have no idea | 643 | 53.45 | 86 | 7.15 | |
| (5) Already have glaucoma | 150 | 12.47 | 191 | 15.88 | |
| Compare with people with your age, do you think you have a higher risk of developing glaucoma? | | | | | |
| (1) Higher | 72 | 5.99 | | | |
| (2) Same | 47 | 3.91 | | | |
| (3) Lower | 275 | 22.86 | | | |
| (4) Have no idea | 659 | 54.78 | | | |
| (5) Already have glaucoma | 150 | 12.47 | | | |

respondents were not able to evaluate their glaucoma risk (53.45%). Physicians were relatively likely to respond with 'unable to evaluate' if the patient was visiting them for the first time. Only 7.15% of the responses from physicians belong to this category; the majority of the responses belong to low risk. In cases when the patients were not able to evaluate their risks but the physician rated them in the low-risk category, the situation was deemed to be of less concern than if the physician had rated them under the high-risk category.

Table 2 shows the weighted sample stratified by whether the patients were in the "High concern" group. Age, sex, education, and marital status were not statistically significant between the two groups. Proportions of correct answers to certain knowledge questions were significantly different between the two groups. Family history of glaucoma and habits of glaucoma checkup were dropped during the model-building process. We retained the age variable despite its non-significance at this stage due to its strong scientific relevancy suggested by previous studies. We believed including this one additional variable would not significantly compromise the efficiency of the model, and it would provide useful information.

Table 3 shows the base model and final model for the logistic regression with dependent = 1 for any respondent in the "High concern" group. Age, sex, education, and marital status were significant predictors in the base model. In this model, knowledge score showed a protective effect. However, in the full model, the estimates for this variable turned non-significant. This shows the significance of adding interaction terms. High levels of education did not exhibit a protective effect; highly educated patients showed relatively high odds of being in the "High concern" group. To assist in the interpretation of the main effect and interaction terms, we calculated the marginal effects of education and sex at each category of age.

Fig 1A–1C show the marginal effects of education, sex, and their interaction effects at different age intervals. Fig 1A shows the marginal effect of knowledge score by education at each age interval. Knowledge score clearly showed a more positive effect on people with college education or above at all age intervals, and this effect was even more significant with increasing age.

Fig 1B shows the marginal effect of knowledge score by sex at each age interval. Despite the overall interaction not reaching statistical significance ($P = 0.202$), for certain age groups, the effect of knowledge showed a clear sex-based disparity. For those in the middle-age groups, women had a clear advantage over men in terms of the effect of knowledge on the outcome variable. Finally, Fig 1C shows the effect of knowledge score by each combination of education and sex at various age intervals. For both men and women, knowledge showed the highest effect for people with the highest level of education. However, women showed a clear advantage over men for people in the highest education group.

**Table 2. Weighted sample characteristic (n = 1203).**

| | Not in high concern group | Row proportion | High concern group | Row proportion | Pearson/Wald test |
|---|---|---|---|---|---|
| | (n = 1062) | | (n = 141) | | |
| | n | | n | | p-value |
| Age (mean, SD) | 54.39 (3.2) | | 53 | 3.42(52.7) | 0.720 |
| Sex (male) | | 0.978 | 51 | 0.022 | 0.068 |
| Education | | | | | |
| Primary school/illiterate | 263 | 0.960 | 30 | 0.040 | 0.089 |
| Junior high school | 124 | 0.990 | 16 | 0.010 | |
| Senior high school | 279 | 0.969 | 38 | 0.031 | |
| College or above | 396 | 0.902 | 57 | 0.098 | |
| Marital status | | | | | 0.059 |
| Never married | 169 | 0.859 | 21 | 0.142 | |
| Married/cohabitate | 786 | 0.974 | 105 | 0.026 | |
| Widowed/separated | 107 | 0.942 | 15 | 0.058 | |
| Self-rated financial status | | | | | |
| Surplus | 394 | 0.960 | 60 | 0.040 | 0.302 |
| Balanced | 568 | 0.926 | 72 | 0.075 | |
| Shortage | 100 | 0.985 | 9 | 0.015 | |
| Glaucoma knowledge by item (correct score) | | | | | |
| 1.Glaucoma can be cured | 275 | 0.971 | 38 | 0.030 | 0.278 |
| 2.Cataract can lead to glaucoma | 196 | 0.977 | 32 | 0.023 | |
| 3.Over use of eyes can lead to glaucoma | 110 | 0.917 | 11 | 0.083 | |
| 4.People with no family history have lower chances of getting glaucoma | 397 | 0.959 | 56 | 0.041 | |
| 5.Early-stage glaucoma does not have symptoms | 290 | 0.968 | 32 | 0.032 | 0.385 |
| 6.Generally speaking, ocular pressure lower than 30mmHg is considered normal. | 135 | 0.956 | 22 | 0.044 | 0.692 |
| Glaucoma symptoms | | | | | |
| 1.Blurred vision | 236 | 0.984 | 27 | 0.016 | 0.025 |
| 2.Pain in the eye | 138 | 0.980 | 16 | 0.020 | 0.146 |
| 3.Halo around light | 115 | 0.980 | 13 | 0.020 | 0.164 |
| 4.Nausea | 73 | 0.992 | 6 | 0.008 | 0.015 |
| 5.Headache | 108 | 0.988 | 11 | 0.012 | 0.040 |
| Glaucoma Knowledge total score (mean, SD) | 1.974(0.4) | | 0.819 (0.5) | | 0.079 |
| Total outpatient times during the past 6 months (mean, SD) | 1.890(0.2) | | 1.395(2.3) | | 0.142 |
| Family history of glaucoma (yes) | 64 | 0.938 | 12 | 0.062 | 0.912 |
| Habit of glaucoma test | | | | | |
| Once per year | 41 | 0.922 | 6 | 0.078 | 0.587 |
| Once per 0.5 year | 82 | 0.972 | 10 | 0.029 | |
| No specific habit | 939 | 0.942 | 125 | 0.058 | |

SD = Standard deviation.

## Discussions

In this study, we examined self-rated risks for developing glaucoma and evaluated this rating against that rated by the physician. We found a considerable proportion of the subjects in the "High concern" group (approximately 12% in our sample). Attaining reasonable glaucoma

**Table 3. Logistic regression for factors associated with high risk, weighted.**

| | Base model | | | | Full model with interactions | | | |
|---|---|---|---|---|---|---|---|---|
| | Odds ratio | 95%CI | | *P* value | Odds ratio | 95%CI | | *P*-value |
| Total glaucoma knowledge score | 0.595 | 0.387 | 0.915 | 0.018 | 1.706 | 0.604 | 4.821 | 0.313 |
| Age | 1.043 | 1.013 | 1.074 | 0.005 | 1.034 | 1.003 | 1.066 | 0.031 |
| Sex (female) | 3.955 | 1.595 | 9.808 | 0.003 | 4.972 | 1.567 | 15.774 | 0.007 |
| Education | | | | | | | | |
| Primary school/illiterate | 1.000 | | | | | | | |
| Junior high school | 0.146 | 0.028 | 0.776 | 0.024 | 0.114 | 0.025 | 0.514 | 0.005 |
| Senior high school | 1.264 | 0.374 | 4.270 | 0.706 | 1.054 | 0.296 | 3.744 | 0.936 |
| College or above | 8.133 | 1.575 | 41.99 | 0.012 | 18.881 | 4.257 | 83.755 | 0.000 |
| Marital status | | | | | | | | |
| Never married | 1.000 | | | | | | | |
| Married/cohabitate | 0.096 | 0.024 | 0.382 | 0.001 | 0.121 | 0.038 | 0.388 | 0.000 |
| Widowed/separated | 0.181 | 0.026 | 1.267 | 0.085 | 0.208 | 0.037 | 1.181 | 0.076 |
| Education*score[1] | | | | | | | | |
| Junior high school | | | | | 1.797 | 0.998 | 3.236 | 0.051 |
| Senior high school | | | | | 0.822 | 0.517 | 1.308 | 0.408 |
| College or above | | | | | 0.294 | 0.136 | 0.634 | 0.002 |
| Sex*score | | | | | | | | |
| Female | | | | | 0.754 | 0.488 | 1.164 | 0.202 |
| Score*age | | | | | 0.999 | 0.985 | 1.014 | 0.931 |
| Goodness-of-fit (p-value) | 0.211 | | | | 0.921 | | | |

[1]Glaucoma knowledge score.

knowledge was protective in terms of inaccurate rating; however, this effect only benefited women and those with high education. The unique niche of this study is the fact that it has explored not only the self-evaluated risk, but also discrepancies of such evaluations with professional ratings. Another contribution of this study is that it explores beyond mere glaucoma knowledge; it works through other critical variables, such as education, age, and sex.

Our results should be discussed in light of the study limitations. First, because we could not accurately calculate the response rate, we did not adjust for non-responses in our weighting. Second, it should be noted that the way we formulated the knowledge score assumed that each knowledge question would have equal weights. In addition, various aspects of glaucoma knowledge may be relevant. Despite selecting the knowledge questions carefully based on the literature and expert opinions, it should be noted that conclusion of this study may depend on the choice of the knowledge measurements.

We found that education and glaucoma knowledge score worked independently on accurate ratings in the base model even when controlled for education. Previous studies have rarely distinguished the two parameters, and in cases where education has been investigated, glaucoma-specific education has been emphasized [15–18]. We argue that education level *per se*, rather than education regarding glaucoma knowledge, plays a significant role in the accuracy of risk rating. Few studies have been published on glaucoma and socioeconomic status; the studies that have been published have not reached a consensus; socioeconomically disadvantaged populations have been found to be associated with high or low prevalence levels of glaucoma, depending on the target population and study design [12, 19, 20]. To add to the existing evidence, we found that such socioeconomic disparity may be explained by its interaction effect with glaucoma knowledge. One plausible explanation is that people with high levels of

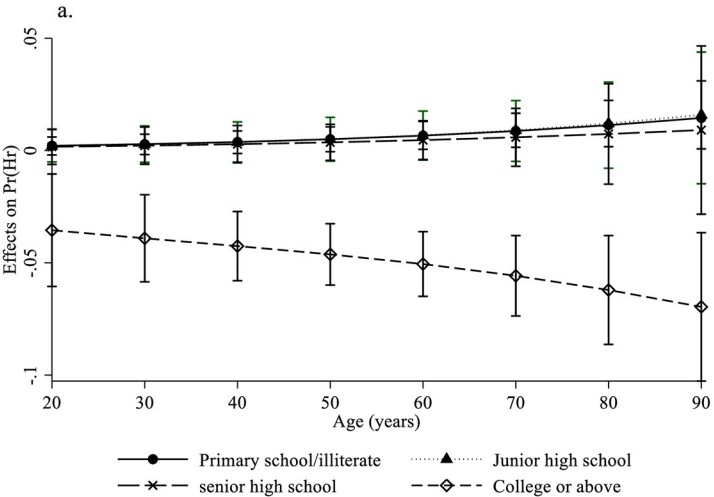

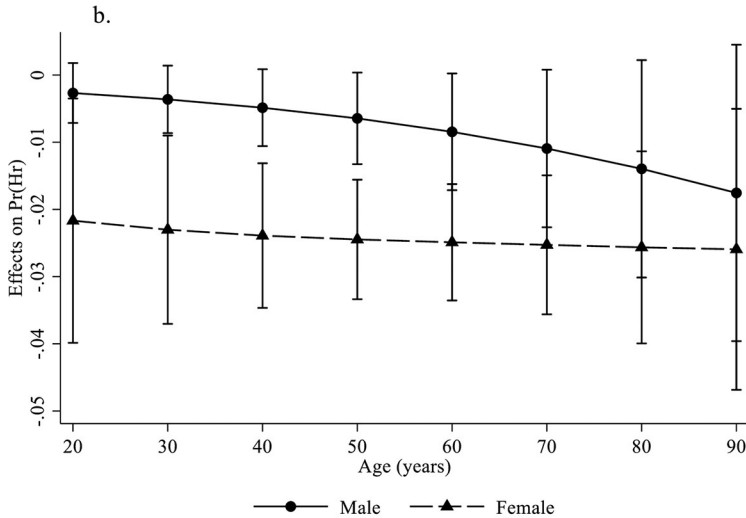

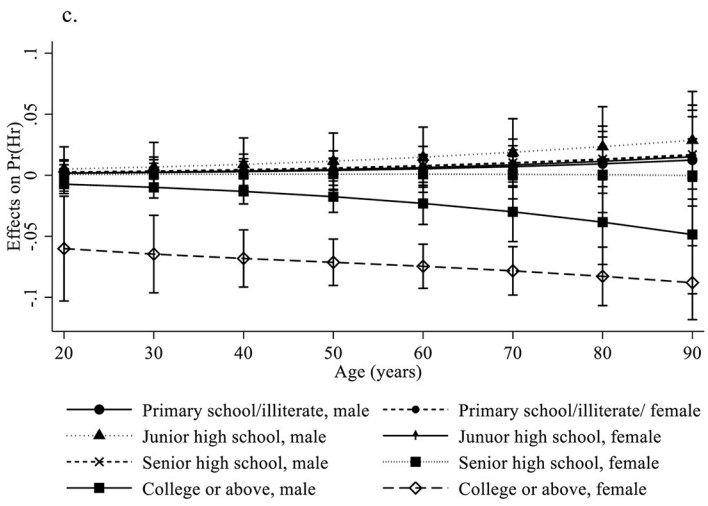

**Fig 1. Marginal effects of glaucoma knowledge.** a. Shows the average marginal effect of glaucoma knowledge score by education at various age intervals. b. Shows the average marginal effect of glaucoma knowledge score by sex at various age intervals. c. Shows the effect of glaucoma knowledge by each education/sex stratum. The marginal effects for Fig 1A–1C should be interpreted as the effect on the predicted probability of being in the "High concern group" in percentage points (point estimate × 100).

education may be able to utilize knowledge effectively and make appropriate judgments regarding the risks. Significant evidence suggests that people with high socioeconomic status can accurately rate their general health evaluated in terms of a robust objective measure, such as mortality [21–23]. However, we also found that the main effect of education follows a reverse trend in favor of the relatively uneducated in both the base and final models. This may suggest that highly educated people with no specific knowledge regarding glaucoma may be overly confident in their health. For example, they are in general more likely to be free of other chronic diseases that are known to have a socioeconomic gradient [24, 25]. By contrast, glaucoma, as mentioned previously, is not associated with a clear socioeconomic gradient.

We also found that the glaucoma knowledge score was associated with a positive effect on reducing the predicted probability of misjudgment. However, the estimates reversed in direction and became non-significant after interaction terms had been added, which has implications both statistically and scientifically. First, the reversal of the estimates from the base model to the full model suggests that disregarding the interactions is not statistically desirable as it leads to reduced model fit. Scientifically, disregarding the interaction terms is associated with misleading results as higher glaucoma knowledge facilitates superior risk assessment that is compatible with conventional wisdom. However, such an effect works with only certain groups of individuals (higher education and female sex in our study).

Age and sex as risk factors have been frequently mentioned in the literature of glaucoma [26–28]. Our study adds to the literature by suggesting that knowledge of glaucoma could be one mechanism in addition to the biological characteristics. Allans et al [29] found that women with a known acute coronary syndrome are more likely to perceive the symptoms as not urgent than are men, thus leading to delay in medical treatment. This finding is in agreement with the main effect of sex in our model. However, our study suggests that women, when equipped with reasonable glaucoma knowledge, respond with more accurate risk perceptions. This sex gap can be closed by glaucoma-specific education. This is an expected trend as marginal effects tend to be larger when baseline effect is smaller, as is the case for the women in our study. In terms of age, glaucoma knowledge has a higher protective effect at an older age, stratified by either sex or education. Advanced age is a risk factor for misjudgment, which warrants policy concern as glaucoma is more prevalent at an older age; thus, misjudgment of glaucoma risk at an advanced age is more likely to lead to late presentation. Glaucoma knowledge is positively associated with high age as found in our study. This disparity also can be reduced by emphasizing glaucoma education. However, as our results suggest, such glaucoma-specific education may positively affect those with high education levels and women.

## Conclusion

A considerable proportion of patients attending the ophthalmology department underestimate their glaucoma risks. Glaucoma-specific knowledge interacts with education and sex, and thus to reduce disparities in inaccurate glaucoma risk perceptions among various education and sex groups, a higher dosage of glaucoma-specific knowledge must be provided to men and to patients with relatively low education levels.

## Supporting information

**S1 File. Patient questionnaire (original language).** Questionnaire used in the current study. (PDF)

**S2 File. Patient questionnaire (English).** Questionnaire used in the current study. (DOCX)

## Author Contributions

**Conceptualization:** Chiun-Ho Hou, Christy Pu.

**Data curation:** Chiun-Ho Hou, Jiahn-Shing Lee, Ken-Kuo Lin, Laura Liu, Yung-Sung Lee, Christy Pu.

**Formal analysis:** Christy Pu.

**Funding acquisition:** Christy Pu.

**Investigation:** Ken-Kuo Lin, Christy Pu.

**Methodology:** Ken-Kuo Lin, Laura Liu, Christy Pu.

**Project administration:** Christy Pu.

**Resources:** Chiun-Ho Hou, Jiahn-Shing Lee, Yung-Sung Lee, Christy Pu.

**Software:** Christy Pu.

**Supervision:** Chiun-Ho Hou, Christy Pu.

**Validation:** Chiun-Ho Hou, Christy Pu.

**Visualization:** Christy Pu.

**Writing – original draft:** Chiun-Ho Hou, Ken-Kuo Lin, Christy Pu.

**Writing – review & editing:** Chiun-Ho Hou, Jiahn-Shing Lee, Ken-Kuo Lin, Laura Liu, Yung-Sung Lee, Christy Pu.

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
