## [Decision Letter · Decision Letter 0]

30 Jul 2021

PONE-D-21-05901

Accuracy of perceived glaucoma risk by patients in a clinical setting

PLOS ONE

Dear Dr. Pu,

Thank you for submitting your manuscript to PLOS ONE. After careful consideration, we feel that it has merit but does not fully meet PLOS ONE’s publication criteria as it currently stands. Therefore, we invite you to submit a revised version of the manuscript that addresses the points raised during the review process.

Please specify the restrictions that do not allow to make the data available.

We look forward to receiving your revised manuscript.

Kind regards,

Timo Eppig

Academic Editor

PLOS ONE

Journal Requirements:

3. Please include in your Methods section (or in Supplementary Information files) the participating hospitals/institutions.

Please include additional information regarding the survey or questionnaire used in the study and ensure that you have provided sufficient details that others could replicate the analyses. For instance, if you developed a questionnaire as part of this study and it is not under a copyright more restrictive than CC-BY, please include a copy, in both the original language and English, as Supporting Information.

Please provide additional information regarding the participant eligibility criteria used and how participants were recruited for the study. 

5. Please include a copy of Table 1b which you refer to in your text on page 10.

Reviewers' comments:

Reviewer's Responses to Questions

**Comments to the Author**

1. Is the manuscript technically sound, and do the data support the conclusions?

Reviewer #1: Yes

2. Has the statistical analysis been performed appropriately and rigorously? 

Reviewer #1: Yes

3. Have the authors made all data underlying the findings in their manuscript fully available?

Reviewer #1: No

4. Is the manuscript presented in an intelligible fashion and written in standard English?

Reviewer #1: Yes

5. Review Comments to the Author

Reviewer #1: Dear authors,

congratulations to your great work! The data emphazise the impact of education about glaucoma disease in the population.

Only some minor points have to be revised:

table 2: "knowledge"

table 3: "Glaucoma" instead of "galaucoma"

The Figures Legends:

Please describe the figures and their results in more detail in order that the readers will understand the main findings.

6. PLOS authors have the option to publish the peer review history of their article (what does this mean?). If published, this will include your full peer review and any attached files.

Reviewer #1: No

---

## [Author Response · Author response to Decision Letter 0]

10 Aug 2021

Dear Editor and Reviewer,

Thank you for the valuable comments on our manuscript. They have indeed helped us enhance our manuscript. We have revised the manuscript based on these comments. All changes are tracked in a different color.

Reviewer #1: Dear authors,

congratulations to your great work! The data emphazise the impact of education about glaucoma disease in the population.

Only some minor points have to be revised:

table 2: "knowledge"

Response: This has been corrected in the revised version

table 3: "Glaucoma" instead of "galaucoma"

Response: Thank you, this has been corrected in the revised version

The Figures Legends:

Please describe the figures and their results in more detail in order that the readers will understand the main findings.

Response: Thank you. More detailed figure legends have been provided in the revised version.

Journal comments:

Response: Thank you. Formatting has been checked.

Response: The reference list has been checked for accuracy.

3. Please include in your Methods section (or in Supplementary Information files) the participating hospitals/institutions.

Response: The hospital name has been added in the revised version.

Please include additional information regarding the survey or questionnaire used in the study and ensure that you have provided sufficient details that others could replicate the analyses. For instance, if you developed a questionnaire as part of this study and it is not under a copyright more restrictive than CC-BY, please include a copy, in both the original language and English, as Supporting Information.

Response: The questionnaire has been included in this submission.

Please provide additional information regarding the participant eligibility criteria used and how participants were recruited for the study. 

Response: Details on participant eligibility criteria have been provided in the revised version.

Response: Detailed IRB information has been provided in the revised version.

5. Please include a copy of Table 1b which you refer to in your text on page 10.

Response: Sorry about the typo. It should have been “Figure 1b”.

---

## [Editor Report · Decision Letter 1]

2 Sep 2021

Accuracy of perceived glaucoma risk by patients in a clinical setting

PONE-D-21-05901R1

Dear Dr. Pu,

We’re pleased to inform you that your manuscript has been judged scientifically suitable for publication and will be formally accepted for publication once it meets all outstanding technical requirements.

Kind regards,

Timo Eppig

Academic Editor

PLOS ONE
---

## [Editor Report · Acceptance letter]

8 Sep 2021

PONE-D-21-05901R1 

Accuracy of perceived glaucoma risk by patients in a clinical setting 

Dear Dr. Pu:

I'm pleased to inform you that your manuscript has been deemed suitable for publication in PLOS ONE. Congratulations! Your manuscript is now with our production department. 

Kind regards, 

on behalf of

Dr. Timo Eppig 

Academic Editor

PLOS ONE